# Preferential Erosion of Soil Organic Carbon and Fine-Grained Soil Particles—An Analysis of 82 Rainfall Simulations

**Michael Hofbauer** [1,2,*]**, David Kincl** [2,3]**, Jan Vopravil** [2,3]**, David Kabelka** [2,4] **and Petr Vráblík** [5]

1    Institute of Hydrodynamics of the Czech Academy of Sciences, Pod Paťankou 30/5,
      160 00 Praha 6, Czech Republic
2    Department of Pedology and Soil Conservation, Research Institute for Soil and Water Conservation,
      Žabovřeská 250, Zbraslav, 156 27 Praha 5, Czech Republic
3    Department of Landscape and Urban Planning, Faculty of Environmental Sciences, Czech University of Life
      Sciences Prague, Kamýcká 129, Suchdol, 165 00 Praha 6, Czech Republic
4    Department of Agroecosystems, Faculty of Agriculture and Technology, University of South Bohemia in
      České Budějovice, Studentská 1668, 370 05 České Budějovice, Czech Republic
5    Department of Environment, Faculty of Environment, Jan Evangelista Purkyně University in Ústí nad Labem,
      Pasteurova 3632, 400 96 Ústí nad Labem, Czech Republic
*    Correspondence: hofbauer@ih.cas.cz

**Abstract:** Soil erosion by water causes the loss of soil mineral particles and soil organic carbon (SOC). For determining the effectiveness of soil conservation measures on arable land, rainfall simulations are regularly carried out in field trials in the Czech Republic. The objective of this study was to analyse a dataset from 82 rainfall simulations on bare fallow soils, containing information on slope inclination, soil texture, soil bulk density, SOC, and soil loss with respect to the preferential erosion of fine-grained soil particles and the enrichment of SOC in the eroded soil. Each rainfall simulation comprised a first rainfall period of 30 min and a second one of 15 min in duration. The rainfall intensity was 1 mm min$^{-1}$ and the kinetic energy of the raindrops accounted for 8.78 J m$^{-2}$ mm$^{-1}$. Runoff samples were taken to determine the soil loss and SOC enrichment in the eroded material. Regression analyses revealed that on sites with <14% slope inclination, SOC mitigated soil loss in the first rainfall period. On sites with >14% slope inclination, soil loss was driven by preferential erosion of fine-grained particles in the first rainfall period. Low soil loss was generally coupled with high SOC enrichment and vice versa, indicating that preferential erosion of SOC occurred mainly in soils with low erosion susceptibility. In order to prevent erosion of SOC and maintain soil quality, soil conservation measures are important in all soil types.

**Keywords:** soil organic carbon; soil erosion; preferential erosion; enrichment ratio; rainfall simulation





## 1. Introduction

Soil organic carbon (SOC) plays an important role in food security and climate change mitigation by means of carbon sequestration [1]. Moreover, SOC is a key parameter for the physical, chemical, and biological quality of soils [2]. Organic matter acts as a binding agent in the formation of soil aggregates, which in turn protect the associated SOC from decomposition [3,4]. Soil organic carbon stabilises soil aggregates against rainfall-induced disruption [5] and increased SOC content reduces soil erosion [6–8].

Conversely, soil erosion causes a loss of SOC, as both mineral particles and organic matter are exported from their original site. Soil erosion is a selective process with respect to both particle size distribution and SOC content; fine soil particles and organic matter are enriched in the eroded material as compared to the source material [7,9–11]. This means that clay, silt, and SOC are preferentially relocated by soil erosion. This comparable behaviour of fine soil particles and SOC is not random, as SOC content and the amount of fine-grained particles of <63 μm size are positively correlated [7]. It was also found that

86% to 91% of the SOC is associated with soil mineral particles [10] and that 77% of the SOC in cropland soils is stored in the fraction of particles < 20 μm [12]. The enrichment ratio (ER) is a measure for the selective erosional process, where ER > 1 indicates an enrichment and ER < 1 indicates a depletion in the eroded material in relation to the source material [7]. Rainfall simulations revealed for soil particles of <20 μm size an ER in between 1.5 and 2.0, for particles from 20 μm to 63 μm size an ER of approximately 1.0, and for particles of >63 μm size an ER of approximately 0.5 [7]. The organic fractions of <63 μm size had an ER in between 1.5 and 2.0 and the organic fractions of >63 μm size had an ER of in between 1.0 and 1.2 [7]. In another experiment, the carbon ER of eroded sediments was found to range between 1.3 and 4.0 [10] and a meta-analysis revealed an average carbon ER of 1.5 for eroded sediments [13].

Soil erosion is predominantly an issue on agricultural land, where soil loss is about 40 times higher than on forestland and about 20 times higher than on other semi-natural vegetation areas [14]. The estimated average soil loss rate due to erosion by water is 3.6 Mg ha$^{-1}$ yr$^{-1}$ for arable land in the EU [15] and 4.2 Mg ha$^{-1}$ yr$^{-1}$ for agricultural land on a global average, respectively [16]. Estimates of SOC loss due to soil erosion by water showed a global average rate of 193 kg ha$^{-1}$ yr$^{-1}$ for cropland [16] and an EU average rate of 68 kg ha$^{-1}$ yr$^{-1}$ for agricultural land [17]. The reasons for the increased erosion susceptibility of agricultural soils in comparison to forestland soils and semi-naturally vegetated soils are the high quantity and intensity of cultivation activities. Therefore, the soil is mechanically disturbed and often not or only sparsely covered by vegetation and plant litter, exposing it to a high risk of soil erosion. Although some approaches such as reduced tillage, no-tillage, organic farming and cover crop cultivation have been proven to effectively mitigate soil erosion [6,18,19], they tend to have an increased risk of reduced yields [20–22]. Therefore, the adoption of such erosion-mitigating technologies is currently limited and soil erosion on arable land still remains a serious issue, leading additionally to decreases in crop yields [23–25].

Rainfall simulations in the field are a method to experimentally determine organic matter loss due to soil erosion by water [7,26]. In the Czech Republic, rainfall simulations are regularly carried out in different field trials throughout the country [26–30]. A frequent aim of these rainfall simulations is the evaluation of the erosion-mitigating potential of different crop management approaches established on the experimental plots, e.g., cover crop and catch crop cultivation, strip-tillage and no-tillage [26–29]. From such measurements, a comprehensive dataset exists. To date, these data have not been analysed regarding the preferential erosion and enrichment of fine-grained soil particles and SOC. However, such information could provide a useful basis for assessing sediment and SOC dynamics in terms of soil erosion. Preferential erosion and enrichment have an influence on sediment and organic carbon transport processes throughout the landscape. Likewise, they influence the ability to trace sediments and organic carbon in the landscape back to their sites of origin. So-called fingerprinting techniques use a variety of physical and biochemical parameters, i.e., fingerprints, for discriminating between sediment sources [31]. Such parameters are often corrected for differences in particles size distribution and organic carbon content between the source soil and the eroded sediment [32,33]. In order to assess the validity of correction factors, however, a better understanding of erosional processes and more attention to the enrichment and depletion effects of fine particles and SOC are needed [32,33]. The present article has the potential to contribute to the comprehension and consideration of such processes. Moreover, it is aimed at providing a basis for deducing appropriate soil conservation measures on arable land.

The objective of this study was to analyse a dataset from the Czech Republic with respect to both the preferential erosion of fine-grained soil particles and the enrichment of SOC in the eroded material. The research questions were as follows: How do soil texture and SOC content influence the soil loss rate? Is the eroded material enriched in SOC as compared to the source material and if it does so, what influence on the enrichment do the soil loss rate, soil texture, and SOC content have?



In order to exclude vegetation impacts with varying influences on erosional processes, only plots with bare fallow treatment were considered. This approach also allowed for emulating the still widespread practice of leaving the soil uncovered for a certain amount of time throughout the year, thereby exposing it to a high risk of soil erosion.

## 2. Materials and Methods

### 2.1. Sites and Rainfall Simulation

From 2014 to 2020, rainfall simulations were carried out from May to September on 19 inclined sites in the Czech Republic. The geographical location of the sites is presented in Figure 1. For each site, the GPS location, total of rainfall simulations, slope gradient, particles size distribution, texture class, SOC content, and soil bulk density are given in Table 1. In order to exclude vegetation impacts and to emulate the practice of leaving the soil uncovered for a certain amount of time, only rainfall simulations carried out on vegetation-free bare fallow were considered in this study. A total of 82 rainfall simulations was analysed.

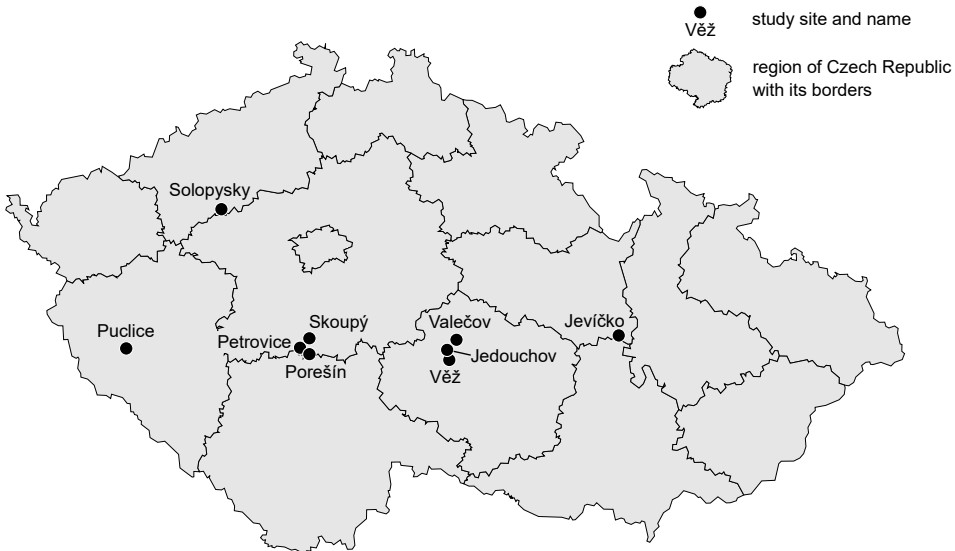

**Figure 1.** Location of the study sites within the Czech Republic. Note: This map is a modified version of the blank vector map provided by mapsvg.com [34], which is distributed under the terms and conditions of the Creative Commons Attribution 4.0 International (CC BY 4.0) license [35].

**Table 1.** Site and soil characteristics of each rainfall simulation site: GPS location, total of rainfall simulations from 2014 to 2020, slope gradient, mean clay (<2 μm) content, mean silt (2 μm to 50 μm) content, mean sand (50 μm to 2000 μm) content, soil texture class according to [36], mean soil organic carbon (SOC) content, and mean soil bulk density.

| Site | GPS Location | Total of Rainfall Simul. | Slope Gradient | Clay Content | Silt Content | Sand Content | Texture Class | SOC Content | Bulk Density |
|---|---|---|---|---|---|---|---|---|---|
| | | | (%) | (g kg⁻¹) | (g kg⁻¹) | (g kg⁻¹) | | (g kg⁻¹) | (g cm⁻³) |
| Jedouchov | 49.5795028N, 15.4489581E | 3 | 12.5 | 86 | 274 | 640 | SL | 13.0 | 1.65 |
| Jevíčko 1 | 49.6257850N, 16.7212519E | 1 | 11.8 | 257 | 490 | 253 | L | 10.3 | (n.d.) [3] |
| Jevíčko 2 | 49.6560531N, 16.7088942E | 10 | 9.3 | 234 | 533 | 234 | SiL | 13.0 | 1.53 |
| Jevíčko 3 | 49.6239792N, 16.7285842E | 3 | 9.4 | 257 | 684 | 59 | SiL | 18.6 | 1.31 |

**Table 1.** *Cont.*

| Site | GPS Location | Total of Rainfall Simul. | Slope Gradient (%) | Clay Content (g kg$^{-1}$) | Silt Content (g kg$^{-1}$) | Sand Content (g kg$^{-1}$) | Texture Class | SOC Content (g kg$^{-1}$) | Bulk Density (g cm$^{-3}$) |
|---|---|---|---|---|---|---|---|---|---|
| Petrovice 1 | 49.5684642N, 14.3257247E | 4 | 13.3 | 181 | 273 | 546 | SL | 10.5 | 1.55 |
| Petrovice 2 | 49.5523536N, 14.3295014E | 8 | 11.7 | 73 | 254 | 674 | SL | 10.7 | 1.44 |
| Petrovice 3 | 49.5617119N, 14.3204194E | 3 | 9.1 | 75 | 214 | 711 | SL | 8.1 | 1.45 |
| Porešín | 49.5379558N, 14.3668522E | 3 | 15.1 | 43 | 255 | 702 | SL | 10.7 | 1.41 |
| Puclice | 49.5864069N, 13.0128503E | 7 | 11.5 | 97 | 460 | 444 | L | 15.5 | 1.44 |
| Skoupý 1 | 49.5798861N, 14.3581947E | 3 | 12.4 | 61 | 365 | 574 | SL | 13.8 | 1.25 |
| Skoupý 2 | 49.5803894N, 14.3593669E | 3 | 13.4 | 61 | 365 | 574 | SL | 15.5 | 1.42 |
| Skoupý 3 | 49.5763589N, 14.3567811E | 2 | 15.6 | 85 | 186 | 729 | SL | 9.4 | 1.57 |
| Solopysky 1 | 50.2591408N, 13.7416328E | 13 | 17.0 | 256 | 444 | 300 | L, CL [1] | 14.2 | 1.49 |
| Solopysky 2 | 50.2562844N, 13.7345892E | 5 | 9.0 | 278 | 432 | 290 | L, CL [2] | 10.7 | 1.38 |
| Valečov | 49.6388550N, 15.4891042E | 3 | 9.6 | 110 | 235 | 655 | SL | 9.7 | 1.39 |
| Věž 1 | 49.5644553N, 15.4517247E | 3 | 8.6 | 129 | 296 | 575 | SL | 9.5 | 1.41 |
| Věž 2 | 49.5751233N, 15.4700092E | 3 | 10.7 | 96 | 322 | 582 | SL | 12.1 | 1.28 |
| Věž 3 | 49.5545625N, 15.4507564E | 2 | 10.2 | 99 | 389 | 512 | L | 10.7 | 1.20 |
| Věž 4 | 49.5629433N, 15.4496219E | 3 | 9.4 | 101 | 410 | 489 | L | 12.8 | 1.42 |

Texture class abbreviations: CL—clay loam; L—loam; SiL—silt loam; SL—sandy loam. [1] For the Solopysky 1 site, the average clay, silt, and sand contents are in accordance with a loam texture. However, 3 out of the 13 samples had a clay loam texture. [2] For the Solopysky 2 site, the average clay, silt, and sand contents are in accordance with a clay loam texture. However, three out of the five samples had a loam texture. [3] (n.d.)—no data.

The rainfall simulator (Figure 2) had four sprinklers with an adjusted water pressure of 50 kPa. The rainfall intensity was set to about 1 mm min$^{-1}$, which corresponds to the average heavy rainfall intensity in the Czech Republic [37]. The average raindrop diameter accounted for 1.5 mm to 2.0 mm and the average kinetic energy of the raindrops accounted for 8.78 J m$^{-2}$ mm$^{-1}$. The sprinklers were installed 2 m above the soil surface and an area of 21 m$^2$ (9.0 m × 2.33 m) was irrigated (Figure 2). The long sides of this area were placed parallel to the runoff direction. In order to prevent lateral runoff, steel sheets bordered the long sides. On the bottom side, the surface runoff was concentrated by means of a flume (Figure 2). A tipping bucket runoff gauge was placed downstream of the flume outlet (Figure 2). The counts of the gauge were recorded by a computer, which allowed for the continuous measurement of surface runoff.

Before each rainfall simulation, the soil was shallowly cultivated and subsequently levelled with a water-filled roller. Each rainfall simulation comprised two rainfall periods. The first rainfall had a duration of 30 min and was carried out at the present soil moisture content in order to simulate a heavy rainfall event on unsaturated soil. After a 15 min break, the second rainfall with a duration of 15 min was performed on the beforehand-irrigated soil in order to simulate a heavy rainfall event on the nearly saturated soil. The break in between both rainfall periods allowed for ponded water for infiltrating or running off until

the start of the second period. For practical reasons, the rainfall simulator was fed with process water used on the farms to which the sites belong, e.g., groundwater or rainwater.

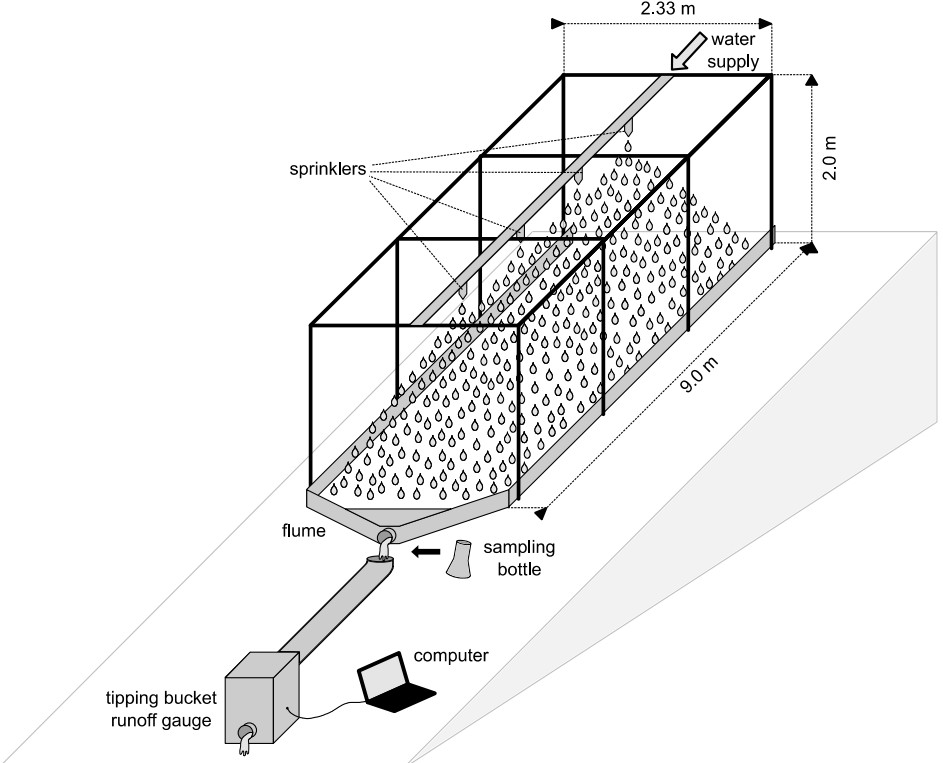

**Figure 2.** Scheme of the rainfall simulator and the equipment setup.

### 2.2. Sampling, Laboratory Analyses, and Calculations

After the surface runoff flow reached the outlet of the flume, runoff samples for collecting the eroded material were taken every three minutes into a sampling bottle with a volume of 319 mL until full (Figure 2). Each runoff sample was decanted into a plastic bottle for transport and stored at room temperature until the analyses. Disturbed soil samples were taken by means of a garden trowel at three points 0.5 m uphill of the irrigation area from 0 cm to 5 cm soil depth, mixed to one bulked sample, and stored in plastic bottles at room temperature until the analyses. Before the start of the rainfall simulation, undisturbed soil core samples (100 cm$^3$ volume) were collected within the irrigation area 0.5 m downhill of its upper boundary from 0 cm to 5 cm soil depth, transported in cooling boxes, and stored in a refrigerator until the analyses.

The runoff samples were dewatered by means of evaporation. The organic carbon content of the source soil (from the disturbed soil samples) and eroded soil (from the runoff samples) was determined by means of sulfochromic oxidation [38]. All of the samples were dried for 12 h at 105 °C in an oven. The particle size distribution of the source soil (from the disturbed soil samples) was analysed by means of sieving and sedimentation [39], and the following texture size classes were determined: clay (<2 μm), silt (2 μm to 50 μm), and sand (50 μm to 2000 μm). Soil bulk density was determined from the undisturbed samples by means of the core method [40]. The weight of the dried eroded sediment (from the runoff samples) was gravimetrically measured.

As runoff samples were taken every three minutes, the sediment concentration in the runoff was calculated in 3 min steps at $k$ times according to:

$$c_i = \frac{m_i}{RO_i}; \; i = 1, 2, \ldots, k, \tag{1}$$

where $c_i$ is the sediment concentration in the runoff at time $i$, $m_i$ is the mass of the eroded sediment at time $i$, and $RO_i$ is the surface runoff at time $i$. Each sediment concentration in the runoff $c_i$ was set constant for three minutes, i.e., until the next runoff sampling time. Runoff data were recorded in 1 s steps for $n$ time intervals and the soil loss for each time interval $j$ was calculated according to:

$$m_j = c_i\, RO_j;\; i = 1,\, 2,\, \ldots,\, k;\; j = 1,\, 2,\, \ldots,\, n;\; k < n, \tag{2}$$

where $m_j$ is the mass of the eroded sediment for the $j$th time interval and $RO_j$ is the surface runoff for the $j$th time interval. The soil loss rate of each rainfall period was calculated according to:

$$\text{soil loss rate} = \frac{\sum_{j=1}^{n} m_j}{A\, t}, \tag{3}$$

where $A$ is the irrigated area, i.e., 21 m$^2$, and $t$ is the duration of rainfall, i.e., 30 min for the first period and 15 min for the second period.

The enrichment ratio of soil organic carbon (ER$_{SOC}$) was calculated according to:

$$\text{ER}_{SOC} = \frac{\text{SOC}_{\text{eroded}}}{\text{SOC}_{\text{source}}}, \tag{4}$$

where SOC$_{\text{eroded}}$ is the soil organic carbon content in the eroded material and SOC$_{\text{source}}$ is the soil organic carbon content in the source soil. Since the ER$_{SOC}$ of the second rainfall period showed a range of remarkably high values (Sections 3.2 and 4.2), these data have been subjected to an outlier detection by means of Tukey's fences ($k = 1.5$). Values with ER$_{SOC} > 3.14$ were recognised as outliers and excluded from the subsequent regression analyses.

### 2.3. Statistical Analyses

Simple linear regression was carried out for modelling the relationships between soil texture, SOC content, soil loss rate, and ER$_{SOC}$. The SOC content in the source soil was fitted against clay (<2 μm) content, silt (2 μm to 50 μm) content, and clay + silt (<50 μm) content; soil loss rate was fitted against clay content, silt content, clay + silt content, and SOC content; and ER$_{SOC}$ was fitted against clay content, silt content, clay + silt content, SOC content, and soil loss rate. To account for the influence of slope steepness on the SOC content, the data were divided into two groups: one with sites with a slope inclination < 14% and one with sites with a slope inclination > 14%, respectively. Simple linear regressions were carried for each of these groups separately. The same procedure was applied for the regressions including ER$_{SOC}$. However, the group of sites with a slope inclination > 14% comprised only eight scattered values, and this was considered as insufficient for drawing a reliable conclusion. Therefore, the regressions for ER$_{SOC}$ are presented for all of the sites together without differentiating according to slope steepness.

Each model was tested for normally distributed residuals by means of the Shapiro–Wilk test. Whenever a linear model did not have normally distributed model residuals, a generalised linear model assuming a Gamma distribution was applied instead. The Gamma distribution was assumed, since the data tended to be right-skewed in these cases. For each parameter fitting combination carried out in the simple linear regression, the Pearson product–moment correlation coefficient ($r$) was calculated.

All of the statistical analyses were carried out by means of the software R, version 4.1.0 (The R Foundation for Statistical Computing, Wien, Austria). The same applied for the creation of the graphs.

## 3. Results

### 3.1. SOC Content in the Eroded Soil and the Soil Loss Rate

The soil organic carbon content in the eroded material had a higher mean and a higher variation in the second rainfall period as compared to the first one (Table 2). The soil loss rate in the second rainfall period was lower than in the first one (Table 2).

### 3.2. Enrichment Ratio of SOC

On average, there was an enrichment of SOC in the mobilised soil as compared to the source soil (Table 2). In the first rainfall period, 68% of the rainfall simulations resulted in an $ER_{SOC} > 1$ and 32% in an $ER_{SOC} < 1$ (Table S1). In the second rainfall period, 67% of the rainfall simulations resulted in an $ER_{SOC} > 1$, 1% in an $ER_{SOC} = 1$, and 32% in an $ER_{SOC} < 1$ (Table S1).

**Table 2.** Mean values for each site and summary statistics across all of the sites for the soil organic carbon (SOC) content in the eroded soil, for the soil loss rate, and for the soil organic carbon enrichment ratio ($ER_{SOC}$), determined in 82 rainfall simulations. The first rainfall period (1st) lasted 30 min and the second rainfall period (2nd) lasted 15 min. For $ER_{SOC}$, the column '2nd ($ER_{SOC} < 3.14$)' contains the mean values and summary statistics of $ER_{SOC}$ in the second rainfall period after removing the data of those 17 rainfall simulations, which resulted in a noticeably high SOC enrichment of $ER_{SOC} > 3.14$ (see Sections 3.2 and 4.2).

| Site | SOC Content in Eroded Soil (g kg$^{-1}$) | | Soil Loss Rate (g m$^{-2}$ min$^{-1}$) | | $ER_{SOC}$ (g kg$^{-1}$/g kg$^{-1}$) | | |
|---|---|---|---|---|---|---|---|
| | Rainfall Period | | Rainfall Period | | Rainfall Period | | |
| | 1st | 2nd | 1st | 2nd | 1st | 2nd | 2nd ($ER_{SOC} < 3.14$) |
| Jedouchov | 19.3 | 14.4 | 14.5 | 23.6 | 1.47 | 1.10 | 1.10 |
| Jevíčko 1 | 11.4 | 10.8 | 9.9 | 24.7 | 1.11 | 1.05 | 1.05 |
| Jevíčko 2 | 12.9 | 12.3 | 31.9 | 46.5 | 1.00 | 0.95 | 0.95 |
| Jevíčko 3 | 20.4 | 36.8 | 22.2 | 13.1 | 1.10 | 1.99 | 1.99 |
| Petrovice 1 | 11.9 | 10.3 | 25.8 | 35.2 | 1.14 | 1.01 | 1.01 |
| Petrovice 2 | 15.2 | 9.6 | 26.1 | 22.3 | 1.48 | 0.92 | 0.92 |
| Petrovice 3 | 14.6 | 9.7 | 19.6 | 24.6 | 1.82 | 1.19 | 1.19 |
| Porešín | 10.0 | 8.9 | 37.8 | 33.1 | 0.94 | 0.85 | 0.85 |
| Puclice | 25.4 | 22.5 | 12.1 | 11.1 | 1.64 | 1.46 | 1.46 |
| Skoupý 1 | 25.8 | 20.4 | 8.1 | 7.3 | 1.91 | 1.49 | 1.49 |
| Skoupý 2 | 25.5 | 23.9 | 2.5 | 4.4 | 1.66 | 1.55 | 1.55 |
| Skoupý 3 | 11.8 | 9.1 | 22.3 | 19.1 | 1.17 | 0.88 | 0.88 |
| Solopysky 1 | 12.3 | 65.5 | 61.6 | 20.6 | 0.88 | 4.62 | 0.63 |
| Solopysky 2 | 14.1 | 37.9 | 19.0 | 8.3 | 1.33 | 3.54 | 1.69 |
| Valečov | 13.4 | 7.9 | 24.3 | 20.1 | 1.37 | 0.82 | 0.82 |
| Věž 1 | 8.6 | 7.6 | 17.2 | 19.6 | 0.91 | 0.78 | 0.78 |
| Věž 2 | 17.0 | 13.9 | 15.0 | 16.7 | 1.44 | 1.17 | 1.17 |
| Věž 3 | 13.3 | 84.9 | 43.8 | 7.4 | 1.25 | 8.11 | (n.a.) [1] |
| Věž 4 | 12.8 | 52.6 | 40.7 | 7.4 | 0.99 | 4.22 | 2.46 |
| median | 13.7 | 13.6 | 21.1 | 14.5 | 1.16 | 1.28 | 1.09 |
| mean | 15.5 | 27.2 | 28.8 | 21.5 | 1.25 | 2.11 | 1.15 |
| std. error | 0.6 | 3.1 | 2.6 | 2.2 | 0.05 | 0.23 | 0.06 |
| min | 4.6 | 2.4 | 1.0 | 2.7 | 0.58 | 0.30 | 0.30 |
| max | 29.0 | 131.4 | 111.7 | 118.8 | 2.35 | 9.48 | 2.80 |
| *n* | 82 | 82 | 82 | 82 | 82 | 82 | 65 |

[1] (n.a.)—not applicable, i.e., at the Věž 3 site, the $ER_{SOC}$ in the second rainfall period was > 3.14 in all cases.

The second rainfall period showed a remarkably higher variation in $ER_{SOC}$ than the first period (Table 2) and 24% of the values exceeded the maximum of the first period (Table S1). The maximum of the second period was about four times higher than the one

of the first period (Table 2). These observations were related to a range of considerably high SOC contents of in between 40.5 g kg$^{-1}$ and 131.4 g kg$^{-1}$ in the eroded material (Table S1). Such high organic carbon contents in the eroded soil were unlikely to be representing typical SOC contents, since the SOC content of the source soil accounted for 6.3 g kg$^{-1}$ to 19.9 g kg$^{-1}$ (Table S1). It was rather supposed that in some rainfall simulations a high quantity of pure organic matter particles was washed out, affecting the measurement of SOC content (Section 4.2). Therefore, Tukey's fences method was carried out and values of ER$_{SOC}$ > 3.14 have been detected as outliers, and this applied to 17 out of 82 rainfall simulations. After removing the data of these 17 rainfall simulations with an ER$_{SOC}$ > 3.14 for further statistical analyses, the ER$_{SOC}$ of the second rainfall period had a similar variation and a similar range like the ER$_{SOC}$ of the first period (Table 2).

### 3.3. Correlations

The soil organic carbon content of the source soil had positive correlations with the silt content, clay + silt content, and to a lesser extent with the clay content (Figure 3).

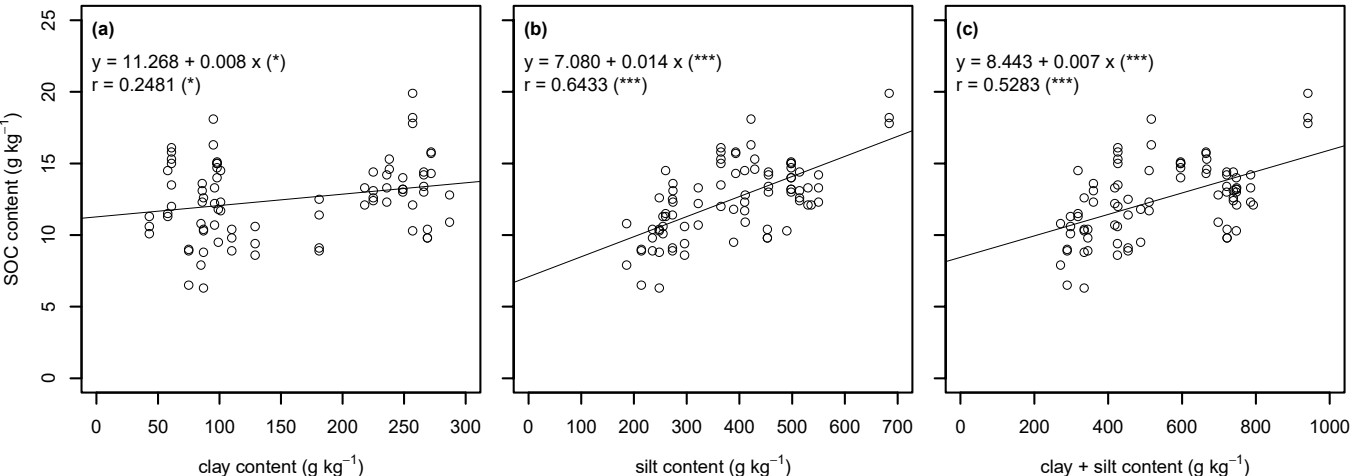

**Figure 3.** Simple linear regression of soil organic carbon (SOC) content against (**a**) clay content, (**b**) silt content, and (**c**) clay + silt content. The regression functions and Pearson product–moment correlation coefficients (*r*) are given for each parameter fitting. The significance levels of the regression function slopes and *r*, indicating a difference from zero, are as follows: (***) *p* < 0.001; (*) *p* < 0.05.

For sites with a slope inclination < 14%, the soil loss correlated negatively with the SOC content in the first rainfall period (Figure 4c) and positively with the clay content in the second rainfall period (Figure 5). For sites with a slope inclination > 14%, the soil loss correlated positively with the silt content, clay + silt content, and SOC content in the first rainfall period (Figure 4).

For the first rainfall period, the ER$_{SOC}$ had negative correlations with the soil loss rate, clay content, clay + silt content, and silt content (Figure 6). For the second rainfall period, the ER$_{SOC}$ showed a negative correlation with the soil loss rate and positive correlations with the SOC content, silt content, and to a lesser extent with the clay + silt content (Figure 7).

Generally, the correlations were not strong as indicated by relatively low Pearson product–moment correlation coefficients *r*, ranging between 0.2354 (ER$_{SOC}$ vs. clay + silt content, Figure 7b) and 0.6433 (SOC content vs. silt content, Figure 3b) for the significant correlations (Table S2).

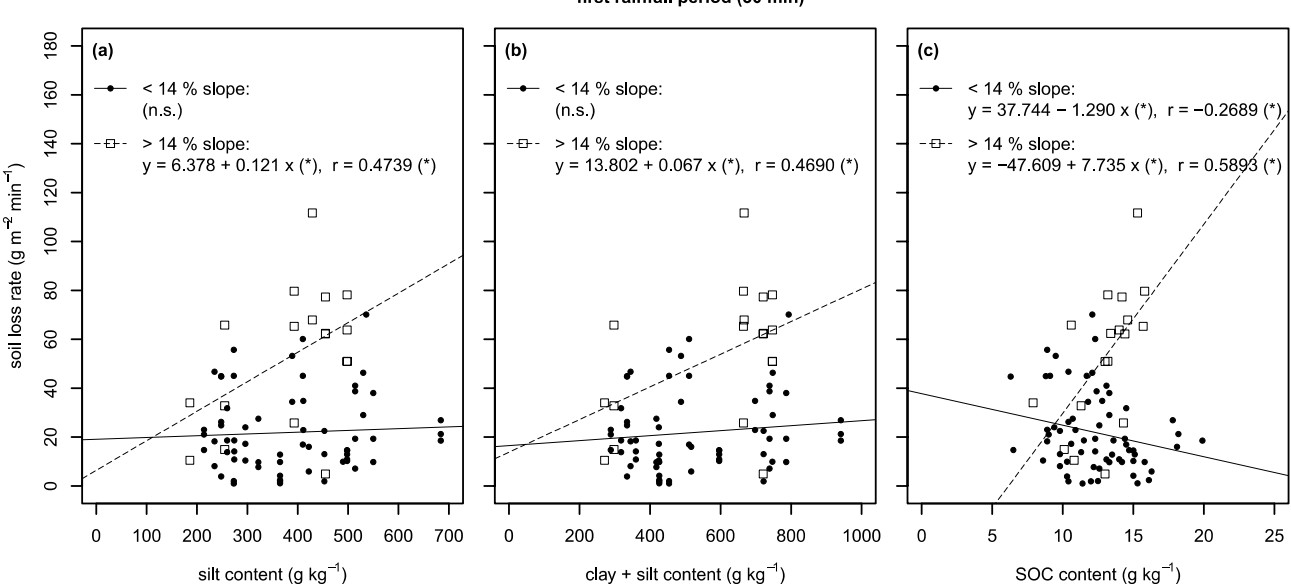

**Figure 4.** Simple linear regression of the soil loss rate in the first rainfall period (30 min) against (**a**) the silt content, (**b**) clay + silt content, and (**c**) soil organic carbon (SOC) content. For the parameter fittings of sites with < 14% slope inclination, generalised linear models assuming a Gamma distribution have been applied. The regression functions and Pearson product–moment correlation coefficients (*r*) are given for the significant parameter fittings. The significance levels of the regression function slopes and *r*, indicating a difference from zero, are as follows: (\*) $p < 0.05$; (n.s.) not significantly different at $p < 0.05$.

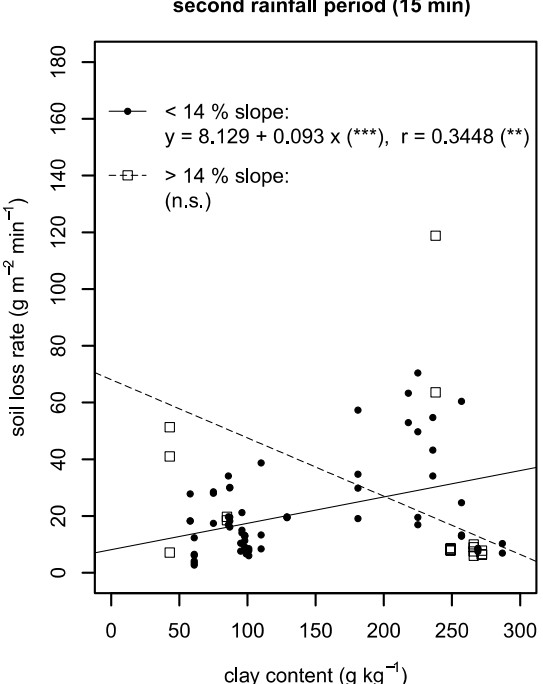

**Figure 5.** Simple linear regression of the soil loss rate in the second rainfall period (15 min) against the clay content. For each parameter fitting, generalised linear models assuming a Gamma distribution have been applied. The regression function and Pearson product–moment correlation coefficient (*r*) are given for the significant parameter fitting. The significance levels of the regression function slope and *r*, indicating a difference from zero, are as follows: (\*\*\*) $p < 0.001$; (\*\*) $p < 0.01$; (n.s.) not significantly different at $p < 0.05$.

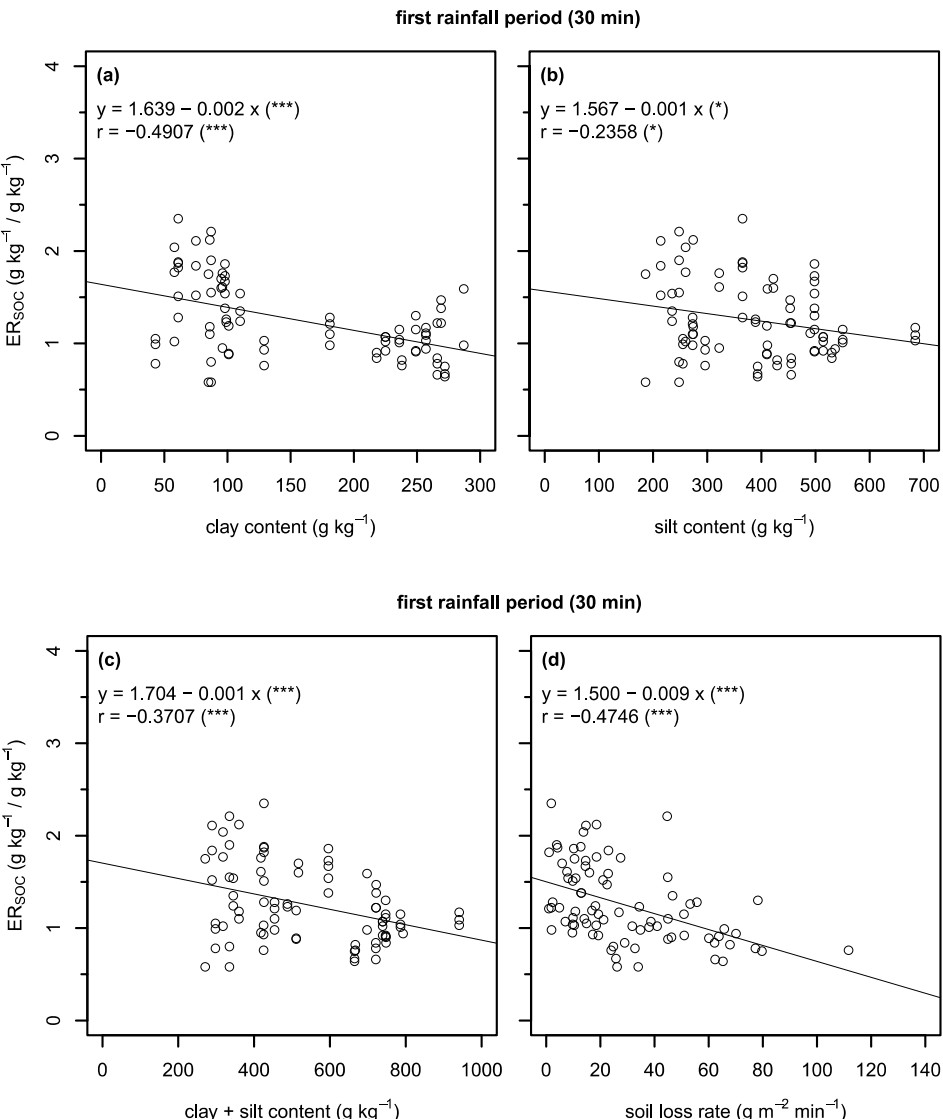

**Figure 6.** Simple linear regression of the soil organic carbon enrichment ratio ($ER_{SOC}$) in the first rainfall period (30 min) against (**a**) the clay content, (**b**) silt content, (**c**) clay + silt content, and (**d**) soil loss rate. The regression functions and Pearson product–moment correlation coefficients (*r*) are given for each parameter fitting. The significance levels of the regression function slopes and *r*, indicating a difference from zero, are as follows: (***) $p < 0.001$; (*) $p < 0.05$.

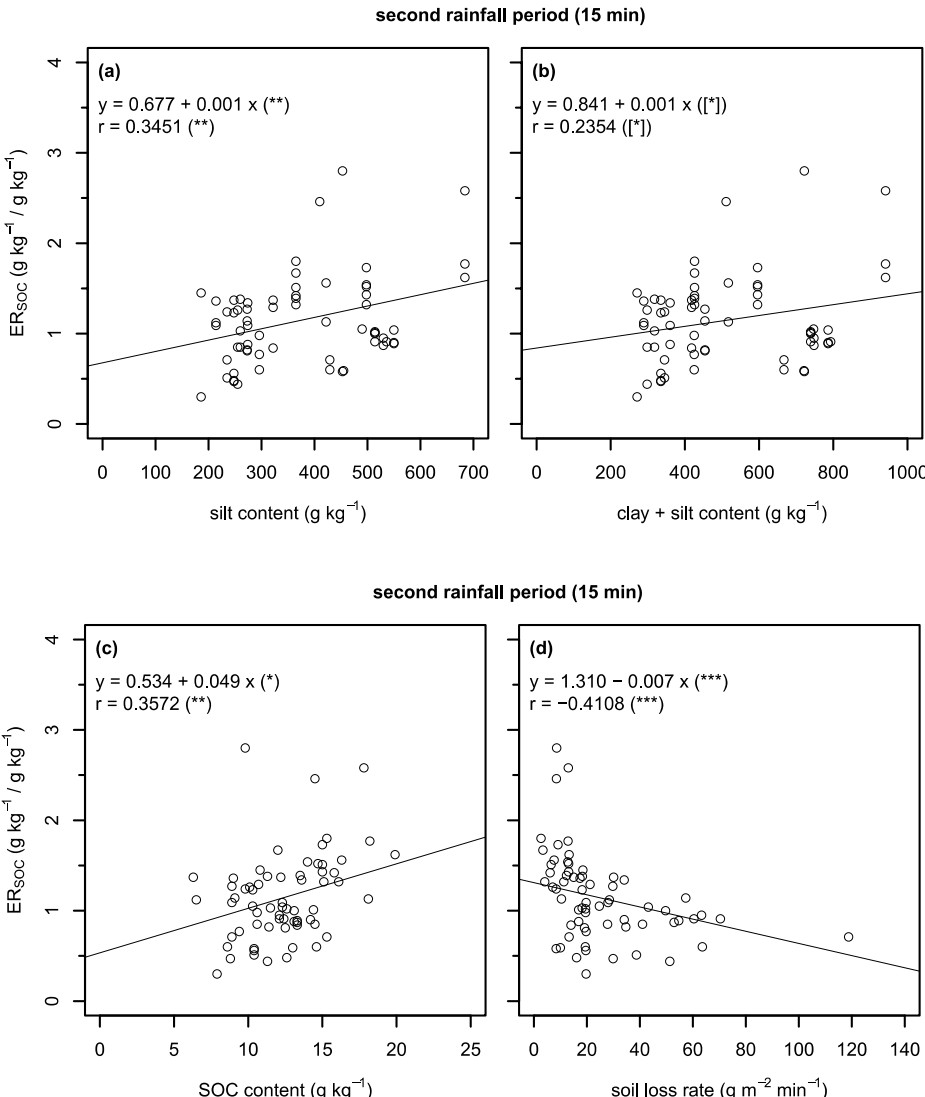

**Figure 7.** Simple linear regression of the soil organic carbon enrichment ratio ($ER_{SOC}$) in the second rainfall period (15 min) against (**a**) the silt content, (**b**) clay + silt content, (**c**) soil organic carbon (SOC) content, and (**d**) soil loss rate. For each parameter fitting, generalised linear models assuming a Gamma distribution have been applied. The regression functions and Pearson product–moment correlation coefficients (*r*) are given for each parameter fitting. The significance levels of the regression function slopes and *r*, indicating a difference from zero, are as follows: (\*\*\*) $p < 0.001$; (\*\*) $p < 0.01$; (\*) $p < 0.05$; ([\*]) $p < 0.1$.

## 4. Discussion

### 4.1. Relationships between the Soil Loss Rate, Soil Texture, and SOC Content

On sites with a slope inclination of >14%, the soil loss rate in the first rainfall period was mainly influenced by the silt content, clay + silt content, and SOC content (Figure 4). On average, the higher the content of these particles, the higher the soil loss rate was. In contrast, no correlation between the soil loss rate and the particle size distribution and SOC content was found for the second rainfall period (Table S2). These observations can be attributed to the preferential mobilisation and transport of fine-grained soil particles and SOC especially at the beginning of the rainfall simulation, resulting in an initial flush of small, easily erodible soil particles in the runoff [7,9,11]. Cheraghi et al. [9] showed that the fine particle (<20 μm) concentration in the runoff quickly rose and reached its peak within the first minutes of precipitation, whereas it started to decrease continuously during the first 20 min of rainfall [9]. Similarly, Asadi et al. [11] found that the contribution

of finer particles to sediment concentration in the runoff was major at the beginning of simulated rainfall, whereas it decreased with ongoing rainfall [11]. Another study revealed an enrichment of particles < 20 μm in size in the eroded material and indicated that the supply of easily erodible particles was limited, since their ER in the 20 min to 40 min rainfall period was smaller than that within the first 20 min of rainfall [7]. Similarly, fine-grained particles, especially silt, were preferentially exported during the first 30 min of rainfall in our experiments, resulting in a depletion of fine particles in the source soil. The particle size distribution has been changed in the course of the first rainfall period and only a few easily erodible particles were present in the soil in the second rainfall period. The original particle size distribution had therefore only little relevance for the soil loss during the second rainfall period, a circumstance that was reflected in the absence of correlations. Regarding the SOC content, a positive correlation with silt content, clay + silt content, and clay content was found (Figure 3). This correlation conforms with other studies, and this is due to the preferential association of SOC with fine mineral particles [7,10,12,41,42]. For this reason, SOC showed a similar behaviour to the fine soil particles, i.e., high SOC content resulted in high soil loss in the first rainfall period, whereas no correlation was found between SOC content and soil loss in the second rainfall period.

On sites with a slope inclination of <14%, the relationships between the soil loss rate, soil texture, and SOC content were different than those on sites with higher slope inclinations. The soil loss rate in the first rainfall period was influenced by the SOC content such that higher SOC content caused lower soil loss (Figure 4). The soil loss rate in the second rainfall period was influenced by the clay content such that the higher the clay content, the higher the soil loss was (Figure 5). From these results it can be deduced that at the beginning of the rainfall simulations, SOC obviously acted as an effective stabiliser against erosion. Numerous other rainfall experiments have shown that higher SOC content reduces soil loss [6–8], as SOC promotes aggregate formation and increases aggregate stability, thereby decreasing the vulnerability of soil aggregates to rainfall-induced disruption [5,7,43]. The results of our experiments suggest that soil aggregates remained quite stable during the first rainfall period, thereby effectively lowering soil loss. With proceeding rainfall, however, the aggregates became gradually instable to such an extent that the erosion-reducing effect of higher SOC content was not effective anymore in the second rainfall period. This aggregate breakdown was possibly related in the first instance to clayey aggregates, since increasing clay content caused an increasing soil loss rate in the second rainfall period, but not in the first one (Figure 5, Table S2). In soils with relatively high clay content, the clay was probably well bound in aggregates and therefore not subject to preferential erosion in the first rainfall period, i.e., there was no initial flush of fine particles which would have caused higher soil loss. However, the wetted aggregates became unstable during the second rainfall period and released clay particles, which were easily erodible and resulted therefore in higher soil loss.

In summary, it seems that on sites with a slope inclination of <14%, soil organic carbon was effective in decreasing soil erosion mainly in the first 30 min of rainfall, probably by means of aggregate formation, which protected the fine soil particles from preferential erosion. On sites with a slope inclination of >14%, soil organic carbon and fine soil particles were preferentially eroded, indicating that SOC and aggregates could not diminish erosion. The reason was probably the steep slope, as it is known that a higher slope inclination causes higher soil loss [7,44].

### 4.2. Enrichment of SOC in the Eroded Material

For the first rainfall period, the enrichment of SOC in the eroded soil was mainly observed when the clay content, silt content, and soil loss rate were low (Figure 6). These results are in accordance with other pieces of literature [7,10,13] and indicate that preferential erosion of SOC was taking place predominantly in soils with relatively high sand contents and relatively low erosion susceptibility. Such coarse-textured soils have a small degree of aggregation and the organic matter in these soils is easily erodible [7,13,45]. This

preferential erosion of SOC in sand-rich soils can be attributed to the shape and the small density of organic particles, both of which require less energy for the mobilisation and transportation of organic matter as compared to mineral particles [7].

In the second rainfall period, 17 considerably high $ER_{SOC}$ values of $3.14 < ER_{SOC} < 9.48$ were found, which were related to the SOC contents in the eroded soil of between $40.5$ g kg$^{-1}$ and $131.4$ g kg$^{-1}$ (Table S1). With respect to the SOC contents in the original soil of only $6.3$ g kg$^{-1}$ to $19.9$ g kg$^{-1}$ (Table S1), it seems unlikely that such a tremendous enrichment of SOC, i.e., of both 'free' particulate organic matter and mineral-associated organic matter, occurred. This assumption is supported by other studies, which showed maximum $ER_{SOC}$ values of approximately 2 [13], 3 [7], and 4 [10], respectively. Moreover, an enrichment with $ER_{SOC} > 3.14$ was found in four sites only (Solopysky 1, Solopysky 2, Věž 3, and Věž 4; Table S1). Therefore, we assume that these high $ER_{SOC}$ values were affected by the washout of a high quantity of coarse particulate organic matter, e.g., plant residues. Such concentrations of organic matter were site-specific and were probably located within the top centimetres of the soil. In the course of the first rainfall period, overlying mineral soil particles were eroded, whereas the organic matter hotspots still remained on the site. During the second rainfall period, this organic matter was exposed on the soil surface and due to ongoing soil erosion, it was finally exported from the site. However, this suggestion needs to be proofed in future research.

When excluding the high values of $ER_{SOC} > 3.14$ from the regression and correlation analyses, the second rainfall period resulted in an enrichment of SOC in the eroded soil also mainly when soil loss rate was low (Figure 7d). This observation is similar to the one in the first rainfall period and underpins the abovementioned finding that the enrichment of SOC occurs especially in soils with low erodibility. Contrary to the first period, however, for coarse-textured soils, there was more SOC depletion and less SOC enrichment in the eroded material (Figure 7b). This circumstance led on average to a smaller $ER_{SOC}$ in the second rainfall period than in the first one (Table 2). The reason for this observation was presumably the preferential erosion of SOC in sandy soils discussed above. The easily erodible fraction of SOC was already washed out during the first rainfall period and the remaining quantity of SOC, which was probably less erodible, was not subjected to preferential erosion in the second rainfall period.

Enrichment of SOC in the eroded soil had a non-significant tendency to decrease with the increasing SOC content in the source soil for the first rainfall period (Table S2), whereas it was positively correlated with the SOC content in the second rainfall period (Figure 7c). In general, the relationship between $ER_{SOC}$ and SOC content was similar to the relationship between $ER_{SOC}$ and fine particle content, especially silt content (Table S2, Figure 7a–c). Such similarity was also found elsewhere [7] and coincides with the strong correlation between SOC content and fine particle content, particularly silt content (Figure 3). Therefore, the influence of SOC content on SOC enrichment seemed to be the result of an association of SOC with fine particles, especially silt. For soils low in SOC, which are typically sandy soils, SOC was mainly not bound to mineral particles and was already easily washed out during the first rainfall period. For soils with higher SOC contents, which are usually finer-textured soils, SOC was bound to mineral particles and eroded together with them, instead of being preferentially washed out.

## 5. Conclusions

The analysis of 82 rainfall simulations on bare fallow soils revealed that on sites with a slope inclination < 14%, soil organic carbon was apparently effective in reducing soil loss in the first phase of rainfall. On sites with >14% slope inclination, however, any probable erosion-mitigating effect of SOC was obviously outclassed by the effect of the steep slope on soil loss. Here, soil loss was mainly driven by preferential erosion of fine-grained particles in the first phase of rainfall. In general, a low soil loss rate was coupled with relatively high enrichment of SOC in the eroded material as compared to the source soil,

indicating that preferential erosion of SOC took place predominantly in soils with low erosion susceptibility.

The results suggest that SOC has some erosion-mitigating potential in bare fallow soils, which is limited by slope steepness, however. Therefore, SOC can be considered as only one element within a wider range of soil conservation measures. This study also showed that enrichment of SOC in the eroded soil depends on soil texture. Lower SOC enrichment in fine-textured eroded soils might be counterbalanced by the higher erosion rates in these soils, thus suggesting similar SOC losses independent of textural class [13]. In order to minimise SOC losses due to soil erosion, the application of soil conservation measures, including permanent soil cover and growing crops with long-term protection against soil erosion [46,47], is therefore important in all soil types.

**Supplementary Materials:** The following supporting information can be downloaded at: https://www.mdpi.com/article/10.3390/agronomy13010217/s1, Table S1: Site characteristics and results for each rainfall simulation; Table S2: Results of regression and correlation analyses.

**Author Contributions:** Conceptualisation: M.H., D.K. (David Kincl) and J.V.; methodology: M.H., D.K. (David Kincl), J.V., D.K. (David Kabelka) and P.V.; software: not applicable; validation: M.H. and D.K. (David Kincl); formal analysis: M.H.; investigation: M.H, D.K. (David Kincl) and D.K. (David Kabelka); resources: D.K. (David Kincl) and J.V.; data curation: M.H. and D.K. (David Kincl); writing—original draft preparation: M.H.; writing—review and editing: D.K. (David Kincl); visualization: M.H.; supervision: D.K. (David Kincl) and J.V.; project administration: D.K. (David Kincl) and J.V.; funding acquisition: D.K. (David Kincl) and J.V. All authors have read and agreed to the published version of the manuscript.

**Funding:** This research was funded by the Ministry of Agriculture of the Czech Republic (Projects QK21020069, QK21010161 and MZE-RO0218) and by the institutional support of the Czech Academy of Sciences (RVO: 67985874).

**Institutional Review Board Statement:** Not applicable.

**Informed Consent Statement:** Not applicable.

**Data Availability Statement:** The data in this study are given in the article and its supplementary materials.

**Acknowledgments:** We sincerely thank all of the partners who supported our field trials and rainfall simulations. We are grateful to Václav Šípek and Jan Hnilica for giving advice during the revision of the manuscript.

**Conflicts of Interest:** The authors declare that they do not have any conflict of interest. The funders had no role in the design of the study, in the collection, analysis, or interpretation of the data, in the writing of the manuscript, or in the decision to publish the results.

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
