# Peer review of "Preferential Erosion of Soil Organic Carbon and Fine-Grained Soil Particles—An Analysis of 82 Rainfall Simulations"

_agronomy, doi:10.3390/agronomy13010217_

Round 1
Reviewer 1 Report (New Reviewer)
This study analyzed the dataset from 82 rainfall simulations on bare fallow soils in the Czech Republic with respect to both a preferential erosion of fine-grained soil particles and enrichment of SOC in the eroded soil, which is timely and relevant research. This supports the needed information to understand better how water erosion processes altered soil carbon dynamics. The manuscript seems to be well-written, and the flow of the story is simple and easy to understand. However, at the moment, the paper is needed to polish, both in terms of data processing, language quality and the statistical analyses. Some of the methods lack any minimal details in order to understand how a given analysis was conducted. I have some comments which may help the authors improve their manuscript. Please, see general and specific comments below.
1、 There are 7 Keywords in your study, which is beyond the normal standard. Please list 3 or 5 most important keywords.
2、 Line 103-109 The description of these 19 inclined sites selected in this study was not so clear, although you listed some GPS location, rainfall simulations and soil texture. However, table 1 in the literature did not list enough information. What are the slope gradient and other soil physical and chemical properties of these of 19 experimental sites?
2、Line 110-118 The experimental design was not so appropriate. As you mentioned in Section 2.1, the sprinklers were installed 2 m above the plot, were they too low? It seems that it is not possible to simulate natural rainfall. And it would be nice if you could provide the experimental schematic diagram or the field picture you took, which is conducive to understand what you conducted.
3、Line 119-126,the first rainfall was carried out on field-moist soil…. What is the field-moist soil? It means the pre-wetting treatment, as other rainfall simulations did?
4、The break time or interval between the first and second rainfall event was 15 min, what’s the principle you conferred?
5、Line 160-175 The statistical methods you applied was very simple, that is to say, simple linear regression. The quality of the articles would go up a notch if you had richer statistical methods.
6、Table 2 I cannot understand how you calculate the soil loss rate. I think the author should be give more explanations and I suggest the authors can supplement some research methods in the Material and Methods, and tell the readers how to compute the soil loss rate in the Table 2.
8、Overall, the Introduction and Discussion part were well organized, but the conclusion was a bit lengthy, therefore I think the authors could make a major revision.
Author Response
Please see the attachment.

Reviewer 2 Report (New Reviewer)
This paper aims to investigate how source soil texture and SOC content influence soil loss rate due to water erosion and also how the source soil texture and SOC content and rate of soil loss affect the enrichment of SOC in the eroded sediments. The manuscript presents work that is very important for the soil erosion and carbon dynamics, and sediment fingerprinting fields. It makes use of a large, unique dataset from the Czech Republic and has the potential to provide valuable information of the relationship between source soil characteristics, soil loss rates and the loss of SOC.
However, the paper currently provides too little information about the dataset, the study sites and why these rainfall simulations have been carried out at these particular sites. The character of the source soils is of vital importance for this research and there is no information provided on how the source soil was collected and stored. More information is required on the sampling procedure, number of replicates, depth of collection etc. This sampling has implications for how comparable the characteristics of the source soil would have expected to be to the sediment being eroded. If the soil samples represent an average of the characteristics in the first 5 to 10 cm of the soil surface how similar would the authors expect this to be to the ~mm of soil surface being eroded during this one rainfall event? Would the authors expect the 1st and 2nd rainfall events to be related to the same layer in the source soil or was the erosion significant enough that the 2nd rainfall event would relate to a deeper layer in the soil? Previous erosion of the field surface and break down of aggregates could have resulted in a crust of finer material that was more easily eroded and would be dissimilar in character to a soil sample taken to a larger depth.
One of my main concerns about the experimental design is the lack of specific site information relating to the topography of each site. Are the slope position and slope angle similar in each case. A difference in slope between sites could result in different soil loss rates and can also affect the accumulation of previously eroded soil. Has this been considered with respect to the results presented.
Abstract/Introduction
The document was well structured but peculiarities in the English used made it slightly hard to read at times. I feel the narrative of the introduction could be better developed to lead more directly to the research questions. A lot of information is provided about the statistics of soil erosion on arable soils but there is no wider discussion of why the arable soils show more erosion than more natural land uses (forest, permanent pasture). i.e. because during times of ploughing, preparation and planting and between crops fields are often left bare/fallow. This is briefly touched on in the last paragraph of the introduction but could be better placed within the body of the text leading to the research questions. The need to link source soil characteristics to SOC erosion dynamics is very important and a dataset of this size could be an invaluable resource. These results therefore have the potential to provide an advancement of the current knowledge however I do not feel that the gap or need for knowledge has been fully identified. One such need is the effect that particle fining and the SOC affinity with finer sediment fractions has on the ability to trace eroded sediments using sediment fingerprinting techniques and trying to disentangle the effects of different land uses and soil types on SOC biomarkers.
More explanation of the existing dataset and the reasons for the data being collected would benefit both abstract and Introduction and may help to crystalise the need and knowledge gap.
Methods
Currently the results would not be reproducible given the current level of detail in the methods section. This section would be greatly improved by the inclusion of a diagram showing the equipment set up, flume and point of collection. It is not clear how the eroded sediment samples were collected from the flume into the “319 ml bottle” – was the bottle put in the flow for a set time or just placed until full? Detail of the source soil collection is required the sampling procedure, number of replicates, spacing of replicates, depth of collection, were they collected directly from the area under the rainfall or nearby etc. Were the eroded sediment samples stored in the bottle, de-watered, stored at room temperature?
Why were the 1st rainfall event (field moist soil) and 2nd rainfall event (~saturated soil) taken over two different time periods (30mins and 15mins respectively)?
With respect to the higher SOC values seen for the 2nd rainfall results – Can these really be considered outliers if they represent ~20% of the measurements?
Have the effects of difference in topography between the different sites been considered. Do the results perhaps need to be presented in groups of similar topographical conditions?
I would have liked to have seen the data (eroded soil SOC content, soil loss rate and ER) from Table S1 within the document rather than as a supplementary table. Perhaps with an average for each site rather than the full information as for Table 1.
Discussion/Conclusion
In general I like the content of the discussion section and the conclusions are consistent with the evidence and arguments presented. However, I do think the results need to be considered more in the light of any differences in the sampling sites e.g. topography, aspect etc. This has been done to some extent for the case of the higher SOC values seen for the 2nd rainfall results [lines 328-337].
Author Response
Please see the attachment.

This manuscript is a resubmission of an earlier submission. The following is a list of the peer review reports and author responses from that submission.
Round 1
Reviewer 1 Report
Regarding Manuscript ID: agronomy-1727843
It seems nice and well-organized paper. However, I would suggest answering some questions and comments as follow:
Please calculate the Fractal dimension of the particle size and investigate the relationship between Fractal and SOC content.
Please show the sampling location on the map.
Please check the effect of the land use on SOC.
I would suggest using the non-linear regression method for the investigation of the relationship between soil particles and SOC.
Reviewer 2 Report
Comments to the reviewed article are included in the attached file.
